# Translocator protein (TSPO) analysis in saliva of adults with oral mucosal lesions: A preliminary study

Guillermo Tamayo-Cabeza[ID][1,2], Valerie Anillo-González[2], Jennifer Orozco-Páez[3], Erika Rodríguez-Cavallo[4], Darío Méndez-Cuadro[5], Farith González-Martínez[2*]

1 Department of Dental Public Health and Dental Informatics, School of Dentistry, Indiana University Indianapolis, Indianapolis, Indiana, United States of America, 2 Department of Research, Public Health Research Group, School of Dentistry, Universidad de Cartagena, Cartagena, Colombia, 3 Analytical Chemistry and Biomedicine Group, Universidad de Cartagena, Cartagena, Colombia, 4 Analytical Chemistry and Biomedicine Group, School of Pharmaceutical Sciences, Universidad de Cartagena, Cartagena, Colombia, 5 Analytical Chemistry and Biomedicine Group, School of Exact and Natural Sciences, Universidad de Cartagena, Cartagena, Colombia

* fgonzalezm1@unicartagena.edu.co

## Abstract

### Background

The analysis of translocator protein (TSPO) in saliva could elucidate its potential role as biomarker in oral mucosal lesions associated to early-stage oral cancer. We compared the TSPO protein signal intensity from saliva samples of adults with and without oral mucosal lesions suspected of being potentially malignant disorders.

### Methods

Saliva samples were collected from 28 participants with and without oral mucosal lesions, recruited from dental clinics in Cartagena, Colombia. A biopsy was performed at the lesion site for each case, and a histopathological diagnosis was obtained. A protein precipitation method for saliva and the Dot blot technique were used to detect and analyze TSPO protein in saliva samples. The signal intensity for TSPO protein was determined by optical densitometry analysis using the software Image Lab. Kruskal-Wallis test was used to compare the intensity of TSPO protein signal by type of histo-pathological diagnosis; while linear regression analysis was used for the association between TSPO protein signal intensity and the presence or absence of oral mucosal lesions, adjusting by age and sex.

### Results

Comparing TSPO protein signal intensity from saliva samples of participants with and without oral mucosal lesions, a higher median was observed in the group of cases with oral mucosal lesions (*p*-value = 0.0141), even after adjusting by age and

**Data availability statement:** All relevant data are within the article and its supporting information files.

**Funding:** This work was supported by the Public Health Research Group, Universidad de Cartagena, Colombia.

**Competing interests:** The authors have declared that no competing interests exist.

sex. TSPO protein signal intensity from saliva samples of participants with dysplasia showed the highest median compared with other histopathological findings ($p$-value = 0.0596).

## Conclusions

A high signal intensity or presence of TSPO protein in saliva samples of participants with oral mucosal lesions may indicate its potential as marker for malignant transformation; therefore, further research should be performed to investigate TSPO expression in oral carcinogenesis.

## Introduction

Oral potentially malignant disorders (OPMDs), previously known as premalignant lesions, are conditions that can lead to oral squamous cell carcinoma (OSCC) [1]. OSCC is the most common oral malignancy of all malignant neoplasms of the mouth, and it is responsible for more than 145,000 deaths worldwide per year [2]. OPMDs consist of leukoplakia, erythroplakia, oral lichen planus, oral submucous fibrosis, and other miscellaneous lesions [3]. Oral histopathological findings such as dysplasia have been linked with the risk of progression to cancer, having a significant transformation rate to cancer that is also related to the grade of dysplasia [4]. Likewise, hyperkeratosis as a histopathological finding can be present in Leukoplakia [5], which is the most common oral potentially malignant disorder [6]. For that reason, the identification and diagnosis of early-stage oral mucosal lesions represent a challenge to reduce the risk of poor prognosis.

Clinical examination and invasive biopsy remain as the standards methods to identify oral cancers; however, the use of biomarkers from saliva samples has been reported to be a promising approach for early diagnosis [7]. Hence, saliva has been extensively studied as a potential source of biomarkers for OSCC, mainly because of its association with the oral environment that can reflect many pathology processes [8]. Among the most studied biomarkers are salivary genomic markers, which are used to identify mutations or alterations in the expression of tumor suppressor genes, including p53, microsatellite instability, abnormal promoter methylation, and the presence of tumor-related viral DNA [9]. Mutations of p53 have been reported to be a reliable and non-invasive alternative for OSCC detection [8].

Several salivary proteins have been investigated as potential biomarkers in human cancers, including oral cancer; among them is the mitochondrial translocator protein (18 kDa), or TSPO—previously known as the peripheral benzodiazepine receptor (PBR) [10–12]. This protein has been detected at various densities in several tissues, and it has been found to be highly expressed at the mitochondrial level of inflammatory cells [13]. TSPO is also found in smaller concentrations in subcellular compartments, on the cell surface as part of the cell membrane, and as a small fraction of the cell nucleus [14].

TSPO is closely associated with the mitochondrial permeability transition pore (mPTP) which relates this protein with the apoptosis regulation and cell death, with ligands that can open the mPTP to induce the apoptosis [15,16]. Nagler and Gavish [17] proposed the 18kDa translocator protein as a salivary biomarker for the diagnosis of oral cancer. Using immunohistochemistry techniques, they analyzed the TSPO expression in oral cancer tumors and reported that increased TSPO levels in oral cancer tissue may be correlated with oral cancer mortality prognosis [12]. However, no studies have been reported on salivary analysis of TSPO in individuals with early-stage oral mucosal lesions or suspected of being potentially malignant.

Because OPMDs can lead to oral cancer progression and transformation, the clinical examination of mucosal lesions suspected of OPMDs in conjunction with the analysis of TSPO in saliva as a potential biomarker may contribute to the evidence concerning the association between TSPO and the identification of early-stage oral cancer lesions. Thus, the present study aimed to compare the TSPO protein signal intensity from saliva samples of adults with and without oral mucosal lesions suspected of being potentially malignant disorders.

## Materials and methods

### Participants

A comparative study was conducted in three centers of stomatological evaluation in Cartagena, Colombia, with a period of recruitment and data collection of three months (September to November 2018). A total of 28 participants were considered eligible for the study: 14 with visual clinical features of OPMDs and 14 participants without oral mucosal lesions. A case was classified as presenting oral mucosal lesion not associated with oral infections and that could originate from various locations within the oral cavity, excluding the oropharynx, soft palate, and the floor of the mouth (including the floor of the tongue). The type of lesions included for histological analysis were white, red, pigmented and lesions without color change such macules, papules, plaques, nodules, clinical hyperplasia, ulcers, erosions, and blisters. Lesions were included if histopathological analysis indicated a diagnosis of hyperplasia, hyperkeratosis, or epithelial dysplasia [18].

Participants were recruited in dental clinics of the stomatology centers of Universidad de Cartagena-School of Dentistry, Hospital Universitario de Cartagena and Hospital Naval de Cartagena, Colombia. Then, participants were matched on sex, age, and origin. Exclusion criteria were as follows: previous history of cancer, chemotherapy and radiotherapy and subjects with hereditary conditions that could increase the risk of cancer.

A questionnaire was used to assess the demographic information (age, sex, and area of residence) and histopathological analysis was performed in the histopathology laboratory group UIBO of the Universidad El Bosque, Bogotá, Colombia. All participants signed a written informed consent form prior the study procedures and authorized the use of their samples for study purpose. Protocols and informed consent forms were reviewed and approved by the Institutional Review Board, School of Dentistry, Universidad de Cartagena (Approval Act Number 002, dated 23/08/2018).

### Histopathological assessment

Biopsy of the tissue was performed for each case by a single oral surgeon in a surgical environment to obtain diagnosis based on pathology reports. A pathologist evaluated all samples where lesions were identified. The samples were classified into three groups: 1) presence of dysplasia (the cells are observed as abnormal: there are cells of different sizes, misshapen cells, intensely pigmented cells and an uncommon number of cells presently dividing, but they are not cancer); 2) presence of epithelial hyperplasia without dysplasia (there is an increase in the number of cells in a tissue that appear normal under a microscope); and 3) presence of acanthosis-hyperkeratosis (diffuse epidermal hyperplasia – associated with a keratin abnormality).

### Saliva samples collection and storage

Each participant provided one saliva sample collected using a pre-chilled 15 mL polypropylene tube. All participants were instructed to abstain from eating and drinking two hours prior to the saliva collection. Saliva samples were collected as

reported elsewhere [19,20] and kept on ice throughout the collection procedure. Then, samples were pretreated adding MilliQ water (1:1 v/v), vortexed for 30 seconds and centrifuged (3500 rpm, 25 min, 4 °C), to obtain one aliquot of 2 mL from the supernatant, and then stored at −40 °C (Fig 1).

## Protein precipitation

A protein precipitation protocol for salivary proteins was employed and adapted, based on the standardized methods reported by Jessie, Hashim (20). The description of this protocol is presented in Fig 1 and reported elsewhere [21]. After obtaining four aliquots during protein precipitation, these were combined to obtain two final aliquots. Protein concentration was determined using the Bradford assay with a calibration curve of bovine serum albumin (BSA) [22].

## Dot blot assay

A dot blot technique was used to detect and analyze TSPO protein in saliva samples. Samples were spotted by duplicate directly onto a nitrocellulose membrane (Amershan™ Protan™ 0.45 μm, catalogue (cat.) #10600002) using a Bio-Dot®

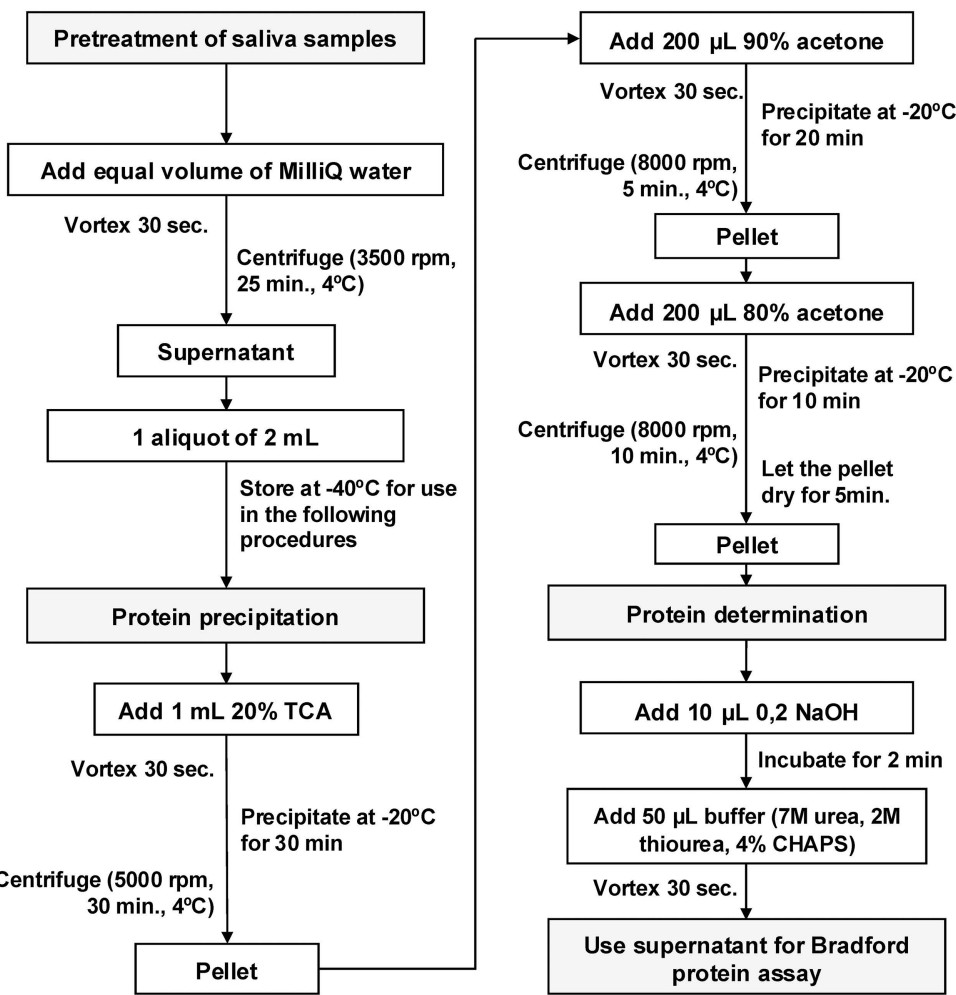

**Fig 1. Protein precipitation protocol for salivary proteins.** Abbreviations: TCA (trichloroacetic acid), CHAPS (3-[(3-Cholamidopropyl) dimethylammonio]-1-propanesulfonate). Source: own elaboration.

microfiltration blotting device (Bio Rad laboratories, Inc., cat. #1706545). Then, the membrane was dried overnight and subsequently blocked with 10% Phosphate Buffered Saline (PBS)-milk for one hour. Upon completion of blocking, the membrane was incubated with primary antibody for two hours at room temperature, using a polyclonal antibody anti-TSPO (EMD Millipore Corporation, cat. #ABC139, dilution 1:2000). Next, the membrane was washed and incubated with secondary antibody Goat anti-Rabbit IgG, HRP (Molecular Probes® Inc., cat #G21234, dilution 1:5000).

Protein spots were visualized in a ChemiDoc™ transilluminator (Bio Rad laboratories, Inc. XRS+ system) by chemi-luminescence reaction using a reagent kit (Invitrogen™ Novex™ ECL Chemiluminescent Substrate Reagent Kit, cat. # WP20005), with a capture protocol of one image each 1.5 minutes for 20 minutes exposition time. The signal intensity for TSPO protein was determined by optical densitometry analysis using the software Image Lab (Bio Rad laboratories, Inc.) to obtain each spot's total intensity with an established area of 15.0 mm$^2$ to keep equals conditions during quantitative analysis (Fig 2 and S1–S3 File).

## Statistical analysis

Data analysis was performed using R version 4.0.2 for Windows [23]. Shapiro Wilk test was used to test for normality on the distribution of numeric variables. Given that the TSPO protein signal intensity did not follow a normal distribution, a non-parametric approach was followed. For categorical variables, number and percentages were obtained. For continuous variables, summary measures were calculated. The signal intensity for TSPO protein in saliva was compared between groups of participants (with and without oral mucosal lesions suspected of being potentially malignant disorders) using the Wilcoxon rank-sum test. Subsequently, the Kruskal-Wallis test was used to compare the TSPO protein signal intensity by type of histopathological diagnosis. Due to the distribution of TSPO signal intensity, a logarithmic transformation was performed on the variable for linear regression to assess the association with the presence or absence of oral mucosal lesions suspected of being potentially malignant disorders, while controlling by age and sex. Values of $p < 0.05$ were considered as indicating statistically significant differences.

## Results

A total of 28 participants were included in the study. The median age was 60.0 years (IQR = 20.3). Regarding sex, 23 (82.0%) participants were females, and 5 (18.0%) were males (Table 1).

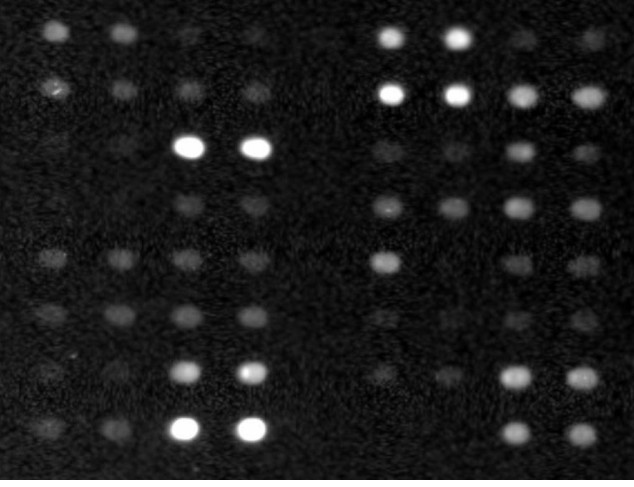

**Fig 2. Salivary TSPO protein Dot blot from donors.** Optical density of each protein spots was measured to determine the protein signal intensity.

**Table 1. Characteristics of participants included in the study.**

| Characteristics | Case (n = 14) | Control (n = 14) | p-value |
|---|---|---|---|
| Age (years), median (IQR) | 61.0 (17.0) | 55 (18.0) | 0.3936[a] |
| Sex, number (percentage) | | | |
| Male | 2 (14.0) | 3 (21.0) | >0.900[b] |
| Female | 12 (86.0) | 11 (79.0) | |
| Histopathological diagnosis, number (percentage) | | | – |
| Dysplasia | 3 (21.4) | – | |
| Epithelial Hyperplasia | 8 (57.2) | – | |
| Acanthosis – Hyperkeratosis | 3 (21.4) | – | |

[a]Wilcoxon rank sum test

[b]Fisher's exact test.

The histopathological diagnoses found in the group of participants with oral mucosal lesions suggested of being OPMDs were dysplasia, epithelial hyperplasia and acanthosis/ hyperkeratosis (Fig 3). Table 1 shows the frequency of these diagnoses. Epithelial Hyperplasia was the most frequent finding (57.2%). The lesions most frequently evaluated were white lesions, although these were more frequent in lesions with dysplasia, and most were plaque lesions. The red lesions were also frequent, and the pigmented lesions were the least frequent.

Comparing the TSPO protein signal intensity from saliva samples by the presence or absence of oral mucosal lesions through optical densitometry analysis, a higher median of signal intensity was observed in participants' samples with oral mucosal lesions (Fig 4A). This difference was found to be statistically significant (p value = 0.0141), even after adjusting by age and sex (Table 2).

TSPO protein signal intensity from saliva samples of participants with dysplasia showed the highest median of signal intensity compared to other histopathological findings (Fig 4B). Conversely, the lowest median of signal intensity was found for acanthosis – hyperkeratosis. However, this difference was not statistically significant (p value = 0.0596).

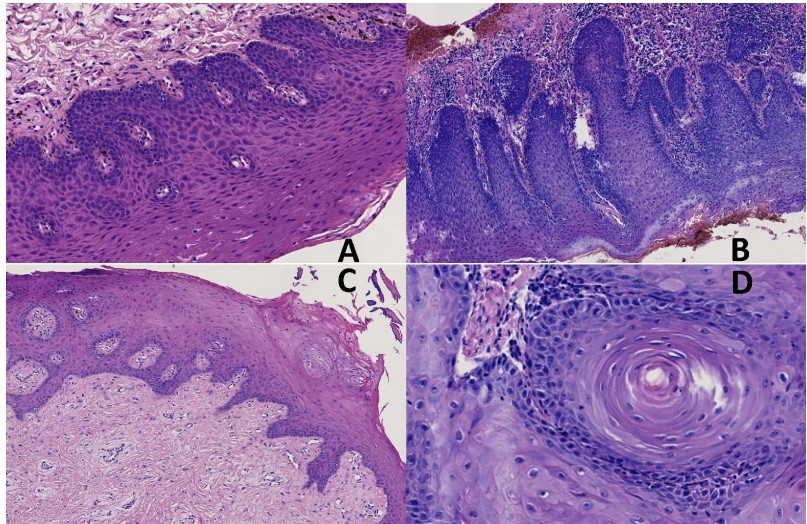

**Fig 3. Histopathological diagnoses found in the group of participants with oral mucosal lesions.** A. Moderate dysplasia. B. Severe dysplasia. C. Epithelial Hyperplasia. D. Hyperkeratosis.

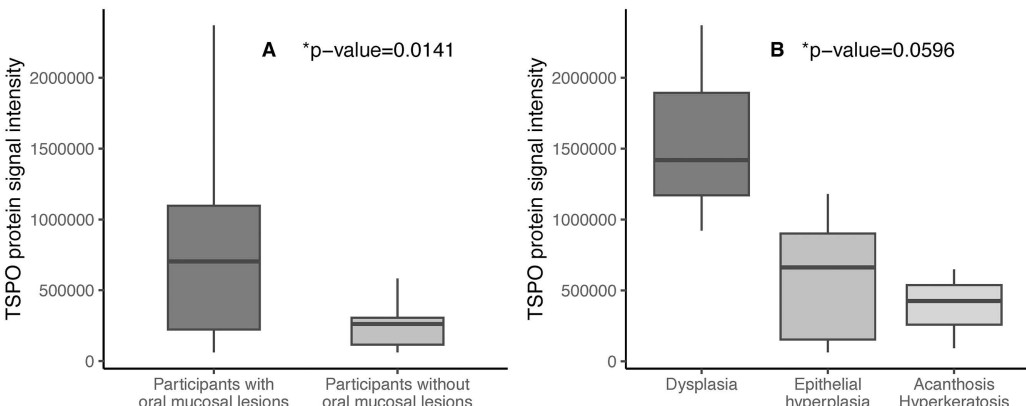

**Fig 4. A.** TSPO protein signal intensity from saliva samples compared by the presence or absence of oral mucosal lesions. *Wilcoxon rank sum test. **B.** TSPO protein signal intensity from saliva samples compared by type of histopathological finding. * Kruskal-Wallis rank sum test.

**Table 2. Linear Regression analysis for the association between TSPO protein signal intensity and the presence or absence of oral mucosal lesions, adjusted by age and sex.**

| Oral mucosal lesions | Estimate (exponentiated) | Standard error | t value | p-value | 95% Confidence interval |
|---|---|---|---|---|---|
| Presence | 2.4073 | 0.3465 | 2.535 | 0.0182* | 1.1774–4.9219 |
| Absence (ref.) | – | – | – | – | – |

Multiple R-squared = 0.2577.

*Wald test.

## Discussion

The use of genes and proteins as biomarkers to assess malignant transformation in oral mucosa lesions has been a common approach in recent decades, often through histopathological analysis of the lesion site [24]. However, these studies can only approximate the molecular events associated with malignant transformation in oral mucosal lesions. Histologically "normal" tissue may not always be "normal" at the molecular level, which introduces potential confounders to the results [25]. Therefore, additional studies are needed to enhance the evidence base, enabling more accurate clinical decision-making.

To our knowledge, this is the first study that compares TSPO protein signal intensity using the Dot blot immunohistochemistry technique in saliva samples of adults with and without oral mucosal lesions. The main findings in this study are the higher presence of TSPO protein observed in saliva samples of participants with oral mucosal lesions suspected of being potentially malignant disorders, among which dysplasia was the type of histopathological finding with the highest TSPO protein signal intensity. This finding, coupled with the absence of reference values for TSPO, encourages further studies aimed at determining its baseline levels in saliva and the thresholds beyond which an increase in its concentration can be associated with the onset of dysplasia. This, along with the noninvasive nature of the test, would give this protein a promising role.

The oral mucosal lesions assessed in the present study were initially considered suspected potentially malignant disorders based on visual clinical examination. However, histopathological assessment revealed that none of the lesions were diagnosed as OPMDs, such as leukoplakia, erythroplakia, oral lichen planus, or oral submucous fibrosis. Instead, abnormalities such as dysplasia, reflecting disordered cell development, and proliferative lesions like epithelial hyperplasia were observed. The presence of dysplasia in histopathologic assessments is considered a predictor for OPMDs' malignant

transformation, and it is an indicator of malignant potential [26]. A meta-analysis of observational studies reported a malignant transformation rate of 10.5% amongst patients with histologically confirmed oral dysplasia undergoing long-term follow up [27].

The lesions with histopathology diagnoses of dysplasia showed the highest median values of TSPO protein signal intensity in the densitometry analysis compared to other histopathological findings, which may indicate a significant presence of TSPO in dysplasia lesions. However, this difference was not statistically significant. It has been suggested that TSPO's overexpression in malignant lesions may be associated with oral carcinogenesis [12]. This association could be explained due to a compensatory attempt of the cell to overcome a binding capacity related malfunctioning [28]. TSPO is involved in regulating apoptotic and necrotic factors in the cytosol as part of the mitochondrial permeability transition pore (mPTP) [29]. Cell death occurs when a prolonged opening of the mPTP results in the release of apoptotic factors such as cytochrome c [30]. The altered regulation of this process may be partially responsible for an unrestrained growth of cancer issues. For that reason, TSPO has attracted attention as a possible molecular marker for cancer.

TSPO protein expression has been reported to correlate positively with disease progression in some cancers, including oral cancer [11,12]. An increase in the expression of TSPO has been observed in other malignant human cells and tissues such as brain tumors [16], prostate cancer [11], pancreatic cancer [31], human thyroid tumors [32] and colon carcinoma [33]. Nevertheless, research of TSPO in the saliva of individuals with oral cancer is limited. Moreover, the use of TSPO as a salivary biomarker in potentially malignant disorders has not been reported yet. Saliva is considered a noninvasive sample and easy collection method for oral cancer detection [7]. Saliva becomes the first choice for screening and identifying biomarkers due to the fallen cancer cells in the oral cavity [34]. Our results suggest the need for further research to study the promising use of TSPO levels in early oral mucosal lesions with carcinogenic potential.

We used the dot blot method because it allows for faster detection of changes in TSPO expression levels and is considered a simplified alternative to the Western blot technique. Although the Dot Blot does not separate the proteins by electrophoresis, its usefulness would be enhanced if it is combined with the Western Blot method because it would allow obtaining information about the size of the protein bands [35]. Therefore, the results of the present study may be limited in identifying modified forms of the target protein. Still, Dot Blots have the advantage of being relatively easy to perform and provide quick, efficient means of examining a range of antibody dilutions or detection substrates [35]. Secondly, our comparative analyzes did not include variables related to smoking habits or cigarettes/tobacco consumption. Smoking remains the most common cause of oral cancer [36], and a relationship between cigarette smoke, TSPO and oral cancer has been suggested [12,28]. Thus, including the frequency of smoking habits could serve to explore the changes of TSPO presence in salivary samples in conjunction with the presence of suspicious oral mucosal lesions. Finally, the relatively small number of cases included in this study compared to other studies also limited the ability to detect differences according to histopathological diagnoses. This was a preliminary study, and a larger sample would be needed to confirm these findings.

This study did not account for conditions such as oral hygiene status or subclinical oral inflammation which may affect salivary protein composition [37]. Although sample collection procedures were standardized and participants refrained from eating or drinking prior to collection, future research should control for these variables to better isolate the association between TSPO expression and oral mucosal lesions.

In summary, our results indicate a higher presence of TSPO protein in saliva samples of participants with oral mucosal lesions compared to participants without oral mucosal lesions, which could be associated to the potential of malignant transformation. Dysplasia was the histopathological finding with the highest TSPO protein signal intensity, compared to epithelial hyperplasia and acanthosis-hyperkeratosis. These results may contribute to the evidence of the association between TSPO and oral carcinogenesis and malignancy, which supports the need of more studies investigating the role of TSPO in the saliva assessment as an oral cancer biomarker.

## Supporting information

**S1 File. Raw dot blot image.**
(TIF)

**S2 File. Inclusivity in global research questionnaire.**
(DOCX)

**S3 File. De-identified data set.**
(XLSX)

## Acknowledgments

We thank the Universidad de Cartagena – School of Dentistry, the Hospital Universitario de Cartagena, and the Hospital Naval de Cartagena for their support in participant recruitment.

## Author contributions

**Conceptualization:** Darío Méndez-Cuadro, Farith González-Martínez.

**Data curation:** Jennifer Orozco-Páez.

**Formal analysis:** Guillermo Tamayo-Cabeza, Jennifer Orozco-Páez.

**Funding acquisition:** Darío Méndez-Cuadro, Farith González-Martínez.

**Investigation:** Guillermo Tamayo-Cabeza, Valerie Anillo-González, Jennifer Orozco-Páez, Erika Rodríguez-Cavallo, Darío Méndez-Cuadro, Farith González-Martínez.

**Methodology:** Jennifer Orozco-Páez, Erika Rodríguez-Cavallo, Darío Méndez-Cuadro, Farith González-Martínez.

**Supervision:** Darío Méndez-Cuadro, Farith González-Martínez.

**Writing – original draft:** Guillermo Tamayo-Cabeza.

**Writing – review & editing:** Guillermo Tamayo-Cabeza, Valerie Anillo-González, Jennifer Orozco-Páez, Erika Rodríguez-Cavallo, Darío Méndez-Cuadro, Farith González-Martínez.

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
