## [Decision Letter · Decision Letter 0]

10 Apr 2025

Dear Dr. Tamayo Cabeza,

Thank you for submitting your manuscript to PLOS ONE. After careful consideration, we feel that it has merit but does not fully meet PLOS ONE’s publication criteria as it currently stands. Therefore, we invite you to submit a revised version of the manuscript that addresses the points raised during the review process.

We look forward to receiving your revised manuscript.

Kind regards,

Anjani Kumar Tiwari, Ph.D.

Academic Editor

PLOS ONE

Journal Requirements:

2. Please include a complete copy of PLOS’ questionnaire on inclusivity in global research in your revised manuscript. Our policy for research in this area aims to improve transparency in the reporting of research performed outside of researchers’ own country or community. The policy applies to researchers who have travelled to a different country to conduct research, research with Indigenous populations or their lands, and research on cultural artefacts. The questionnaire can also be requested at the journal’s discretion for any other submissions, even if these conditions are not met.  

Please find more information on the policy and a link to download a blank copy of the questionnaire here: https://journals.plos.org/plosone/s/best-practices-in-research-reporting. 

Please upload a completed version of your questionnaire as Supporting Information when you resubmit your manuscript.

“This article is a product of research funded by Group of Research in Public Health, Universidad de Cartagena in Colombia.”

Please state what role the funders took in the study.  If the funders had no role, please state: "The funders had no role in study design, data collection and analysis, decision to publish, or preparation of the manuscript.” If this statement is not correct you must amend it as needed. 

4. Please note that funding information should not appear in the Acknowledgments section or other areas of your manuscript. We will only publish funding information present in the Funding Statement section of the online submission form. Please remove any funding-related text from the manuscript. 

5. We note that you have indicated that there are restrictions to data sharing for this study. For studies involving human research participant data or other sensitive data, we encourage authors to share de-identified or anonymized data. However, when data cannot be publicly shared for ethical reasons, we allow authors to make their data sets available upon request. For information on unacceptable data access restrictions, please see http://journals.plos.org/plosone/s/data-availability#loc-unacceptable-data-access-restrictions. 

6. For studies involving third-party data, we encourage authors to share any data specific to their analyses that they can legally distribute. PLOS recognizes, however, that authors may be using third-party data they do not have the rights to share. When third-party data cannot be publicly shared, authors must provide all information necessary for interested researchers to apply to gain access to the data. (https://journals.plos.org/plosone/s/data-availability#loc-acceptable-data-access-restrictions) 

7. PLOS ONE now requires that authors provide the original uncropped and unadjusted images underlying all blot or gel results reported in a submission’s figures or Supporting Information files. This policy and the journal’s other requirements for blot/gel reporting and figure preparation are described in detail at https://journals.plos.org/plosone/s/figures#loc-blot-and-gel-reporting-requirements and https://journals.plos.org/plosone/s/figures#loc-preparing-figures-from-image-files. When you submit your revised manuscript, please ensure that your figures adhere fully to these guidelines and provide the original underlying images for all blot or gel data reported in your submission. See the following link for instructions on providing the original image data: https://journals.plos.org/plosone/s/figures#loc-original-images-for-blots-and-gels.   

**Additional Editor Comments:**

As per the reviewers' recommendations, the manuscript needs to be revised in accordance with their comments before resubmission to PLOS ONE

Reviewers' comments:

Reviewer's Responses to Questions

**Comments to the Author**

1. Is the manuscript technically sound, and do the data support the conclusions?

Reviewer #1: Yes

Reviewer #2: Yes

2. Has the statistical analysis been performed appropriately and rigorously?

Reviewer #1: Yes

Reviewer #2: Yes

3. Have the authors made all data underlying the findings in their manuscript fully available?

Reviewer #1: Yes

Reviewer #2: Yes

4. Is the manuscript presented in an intelligible fashion and written in standard English?

Reviewer #1: No

Reviewer #2: Yes

Reviewer #1: 1. The manuscript present a good clinical study, when I tried to search similar articles, I found DOI: 10.1111/odi.13178 where the authors have done similar work. the authors of the current manuscript may use that information and try to draw some comparison.

2. language is issue so rectify the deficiencies

3. The authors obtained permission in 2018 and submitted in 2024, why so late

4. Image resolution is issue, get these corrected

5. Comparison of TSPO level with a standard is missing try to do some meaning comparison

Reviewer #2: In order to assess the changes in cancer cells, evaluation of biomarker is important. TSPO has been considered as biomarker for cancer. This can predict the changes at molecular level much before other modalities can predict it. The study carried out for correlation of TSPO intensity with potential malignancy could be significantly useful due to non invasive sampling. Clarification on following points are required.

How do we concentrate the various biomarkers taken from the saliva of the patients?

Can a direct correlation be drawn between TSPO signal intensity and presence or absence of oral mucosal lesions ? If yes, then how ?

**Do you want your identity to be public for this peer review?** For information about this choice, including consent withdrawal, please see our Privacy Policy

Reviewer #1: No

Reviewer #2: No

---

## [Author Response · Author response to Decision Letter 1]

16 May 2025

May 16th, 2025

Dear Dr. Anjani Kumar Tiwari

Academic Editor

PLOS ONE

I hope this letter finds you well. As the first author of the article titled “Translocator Protein (TSPO) Analysis in Saliva of Adults with Oral Mucosal Lesions: A Preliminary Study,” submitted to PLOS ONE, I would like to extend my appreciation for the review provided and for the consideration of our work for publication in the journal.

The suggested changes have been reviewed in detail, and below I describe how each of these was incorporated into the revised version submitted through the system:

Editor comment: If applicable, we recommend that you deposit your laboratory protocols in protocols.io to enhance the reproducibility of your results.

Response: We appreciate the suggestion to publish our laboratory protocol in protocols.io. We confirmed that the protocol has been published and it has been assigned the following DOI: dx.doi.org/10.17504/protocols.io.kxygxqxdkv8j/v1

We have also included it as a reference in the revised manuscript (Reference #21).

Editor comment: Please ensure that your manuscript meets PLOS ONE's style requirements, including those for file naming.

Response: The manuscript has been adapted to meet the journal’s style requirement as described in the instructions provided: “Manuscript Body Formatting Guidelines” and “Title, Author, Affiliations Formatting Guidelines.”

Editor comment: Please include a complete copy of PLOS’ questionnaire on inclusivity in global research in your revised manuscript.

Response: A complete copy of PLOS’ questionnaire on inclusivity in global research was submitted to the system as requested and added as a supplementary information in the manuscript.

Editor comment: Please note that funding information should not appear in the Acknowledgments section or other areas of your manuscript.

Response: The funding information has been removed from the Acknowledgements section as requested. This information has been included in the online submission form: “This article is a product of research funded by the Public Health Research Group from Universidad de Cartagena in Colombia. The funders had no role in study design, data collection and analysis, decision to publish, or preparation of the manuscript.”

Editor comment: We note that you have indicated that there are restrictions to data sharing for this study.

Response: The de-identified data set has been submitted as a Supplementary Information File #3. We confirm that the dataset has been anonymized and that the included ID information does not contain any identifiable details about the participants. Additionally, the data is being shared in accordance with local and ethical regulations.

Editor comment: PLOS ONE now requires that authors provide the original uncropped and unadjusted images underlying all blot or gel results reported in a submission’s figures or Supporting Information files.

Response: The original and unadjusted image underlying the salivary TSPO protein dot blot from donors has been submitted as Supplementary file #1 in its original .tif format.

Reviewer #1 comment: The manuscript presents a good clinical study, when I tried to search similar articles, I found DOI: 10.1111/odi.13178 where the authors have done similar work.

Response: We appreciate the positive feedback regarding the study. With respect to the reference of Nagler, Wizman and Gavish (2019), this key study was included as a reference in the original submission. The comparison with their work had also been previously included:

Lines 210 to 215:

“The lesions with histopathology diagnoses of dysplasia showed the highest median values of TSPO protein signal intensity in the densitometry analysis compared to other histopathological findings, which may indicate a significant presence of TSPO in dysplasia lesions. However, this difference was found not statistically significant. It has been suggested that TSPO's overexpression in malignant lesions may be associated with oral carcinogenesis (Nagler et al., 2010). This association could be explained due to a compensatory attempt of the cell to overcome a binding capacity related malfunctioning (Nagler et al., 2019).”

Lines 238 to 241:

“Secondly, our comparative analyzes did not include variables related to smoking habits or cigarettes /tobacco consumption. Smoking remains the most common cause of oral cancer (P. Chaturvedi et al., 2019), and a relationship between cigarette smoke, TSPO and oral cancer has been suggested (Nagler et al. 2010, 2019). Thus, including the frequency of smoking habits could serve to explore the changes of TSPO presence in salivary samples in conjunction with the presence of suspicious oral mucosal lesions.”

Reviewer #1 comment: language is issue so rectify the deficiencies

Response: We appreciate the reviewer’s comment regarding language. We have carefully revised the manuscript to improve clarity and grammar throughout. The edits have been made using Track Changes, so all modifications to the language are clearly visible in the revised document.

Reviewer #1 comment: The authors obtained permission in 2018 and submitted in 2024, why so late.

Response: We acknowledge the delay between data collection and manuscript submission. After completing the laboratory analysis in 2018, the first author began a Master's degree and subsequently a PhD program abroad, which extended the timeline for manuscript preparation. Additionally, time was required for a thorough literature review and coordination with co-authors to ensure the quality and accuracy of the final submission. We confirm that the data remain valid and the study addresses a research question that is still relevant to the field.

Reviewer #1 comment: Image resolution is issue, get these corrected

Response: We have submitted to the journal’s system the images and figures with better resolution (up to 600 pixels/inch) using the .tiff format.

Reviewer #1 comment: Comparison of TSPO level with a standard is missing try to do some meaning comparison

Response: Currently, there are no established reference values for TSPO in blood, saliva, or other body fluids. Baseline values are typically very low and can vary significantly between individuals and under different physiological stress response conditions. In this context, the relative expression levels of this protein were used to identify differential behavior between clinically well-established groups, as a first approximation of its potential role as a biomarker. Therefore, we added the following explanation to the discussion:

Lines 196 to 199:

“This finding, coupled with the absence of reference values for TSPO, encourages further studies aimed at determining its baseline levels in saliva and the thresholds beyond which an increase in its concentration can be associated with the onset of dysplasia. This, along with the noninvasive nature of the test, would give this protein a promising role.”

Reviewer #2 comment: In order to assess the changes in cancer cells, evaluation of biomarker is important...

Response: We thank the reviewer for this positive and encouraging comment. We agree that the non-invasive assessment of TSPO as a potential biomarker holds promise for early molecular detection of malignancy.

Reviewer #2 comment: How do we concentrate the various biomarkers taken from the saliva of the patients?

Response: We thank the reviewer for this question. To concentrate salivary proteins, including TSPO, we applied a protein extraction and precipitation protocol using trichloroacetic acid (TCA) and CHAPS, as adapted from Jessie and Hashim (https://doi.org/10.3923/biotech.2008.686.693). This step allowed for enrichment of total proteins from saliva prior to dot blot analysis. The description of this process is presented in Figure 1, and we have published the protocol (using protocols.io) publicly: dx.doi.org/10.17504/protocols.io.kxygxqxdkv8j/v1

This protocol has been included as reference #21 in the revised manuscript.

Reviewer #2 comment: Can a direct correlation be drawn between TSPO signal intensity and presence or absence of oral mucosal lesions? If yes, then how?

Response: While our results showed higher TSPO signal intensity in saliva samples of participants with oral mucosal lesions compared to those without, this finding should be interpreted in the context of the study’s limited sample size. As a preliminary investigation, we cannot draw a direct correlation at this stage. Rather, our findings highlight a potential association that warrants further investigation in larger, well-controlled studies.

Attached comment: Confounding factors such as oral hygiene, inflammation, or diet

Response: We agree that factors such as inflammation may influence salivary protein levels. In this preliminary study, we attempted to reduce variability by instructing all participants to refrain from eating or drinking two hours prior to sample collection and by processing all samples under standardized laboratory conditions. However, we acknowledge that we did not collect specific data on oral hygiene practices or inflammatory status, and these may represent associated factors. We have now noted this limitation in the discussion in lines 305 to 316 and emphasized the need to control for these variables in future studies.

Lines 248 to 251:

“This study did not account for conditions such as oral hygiene status or subclinical oral inflammation which may affect salivary protein composition [37]. Although sample collection procedures were standardized and participants refrained from eating or drinking prior to collection, future research should control for these variables to better isolate the association between TSPO expression and oral mucosal lesions.”

Attached comment: The sample size of just 28 participants appears to be relatively small...

Response: We appreciate the reviewer’s concern regarding the sample size. As noted throughout the manuscript, this is a preliminary study designed to assess the feasibility of salivary TSPO protein detection in individuals with and without oral mucosal lesions. We acknowledge that the small sample size limits the generalizability of our findings and the statistical power to detect subtle associations. This limitation has been discussed in the manuscript (Lines 243 to 246) and further emphasized the need for larger studies to validate and expand upon these initial observations.

Attached comment: Clinical implications of TSPO detection in oral lesions

Response: While our study found higher TSPO levels in cases of dysplasia, this difference was not statistically significant, and we do not suggest any clinical application at this stage. However, our findings support the feasibility of salivary TSPO detection in patients with oral mucosal lesions and highlight its potential role as a non-invasive biomarker. If validated in larger studies with longitudinal follow-up, TSPO could contribute to earlier identification of lesions at risk for malignant transformation, thereby complementing clinical examination and histopathology.

Attached comment: Methodological rigor of dot blot technique, controls, and sensitivity/specificity

Response: Regarding the absence of analysis of sensitivity and specificity in our study, currently, there are no established reference values for TSPO in blood, saliva, or other body fluids. Baseline values are typically very low and can vary significantly between individuals and under different physiological stress response conditions. In this context, the relative expression levels of this protein were used to identify differential behavior between clinically well-established groups, as a first approximation of its potential role as a biomarker. Therefore, we added the following explanation to the discussion:

Lines 196 to 199:

“This finding, coupled with the absence of reference values for TSPO, encourages further studies aimed at determining its baseline levels in saliva and the thresholds beyond which an increase in its concentration can be associated with the onset of dysplasia. This, along with the noninvasive nature of the test, would give this protein a promising role.”

We appreciate the reviewers’ and editors’ constructive feedback, which has helped strengthen the clarity and quality of our manuscript.

Best regards,

Guillermo Tamayo-Cabeza, BDS, MSc

PhD student in Dental Sciences, Research Assistant

Department of Dental Public Health and Dental Informatics

Indiana University School of Dentistry

gtamayo@iu.edu

---

## [Decision Letter · Decision Letter 1]

28 Jul 2025

Translocator Protein (TSPO) Analysis in Saliva of Adults with Oral Mucosal Lesions: A Preliminary Study

PONE-D-24-51043R1

Dear Dr. Tamayo Cabeza,

We’re pleased to inform you that your manuscript has been judged scientifically suitable for publication and will be formally accepted for publication once it meets all outstanding technical requirements.

Kind regards,

Anjani Kumar Tiwari, Ph.D.

Academic Editor

PLOS ONE

Additional Editor Comments (optional):

I think, manuscript is suitable for publication in PLOS One after corrections suggested by reviewers.

Reviewers' comments:

Reviewer's Responses to Questions

**Comments to the Author**

Reviewer #2: All comments have been addressed

2. Is the manuscript technically sound, and do the data support the conclusions?

Reviewer #2: Yes

3. Has the statistical analysis been performed appropriately and rigorously?

Reviewer #2: (No Response)

4. Have the authors made all data underlying the findings in their manuscript fully available?

Reviewer #2: Yes

5. Is the manuscript presented in an intelligible fashion and written in standard English?

Reviewer #2: Yes

Reviewer #2: (No Response)

**Do you want your identity to be public for this peer review?** For information about this choice, including consent withdrawal, please see our Privacy Policy

Reviewer #2: No

---

## [Editor Report · Acceptance letter]

PONE-D-24-51043R1

PLOS ONE

Dear Dr. Tamayo-Cabeza,

I'm pleased to inform you that your manuscript has been deemed suitable for publication in PLOS ONE. Congratulations! Your manuscript is now being handed over to our production team.

Kind regards,

on behalf of

Dr. Anjani Kumar Tiwari

Academic Editor

PLOS ONE